# One Diffusion Step to Real-World Super-Resolution via Flow Trajectory Distillation

**Jianze Li** [* 1]  **Jiezhang Cao** [* 2]  **Yong Guo** [3]  **Wenbo Li** [4]  **Yulun Zhang** [† 1]

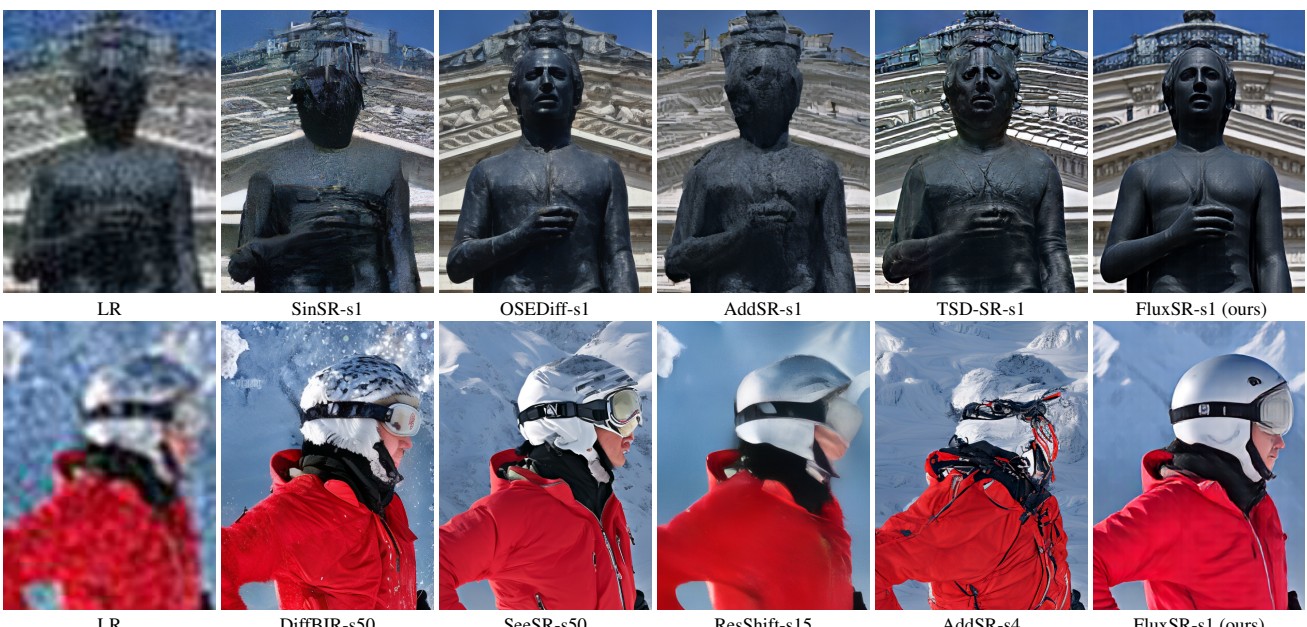

Figure 1: Visual comparisons of different Real-ISR methods. Top: Comparison between FluxSR and state-of-the-art one-step diffusion methods. Bottom: Comparison between FluxSR and state-of-the-art multi-step diffusion methods. Our proposed FluxSR generates more realistic images with high-frequency details.

## Abstract

Diffusion models (DMs) have significantly advanced the development of real-world image super-resolution (Real-ISR), but the computational cost of multi-step diffusion models limits their application. One-step diffusion models generate high-quality images in a one sampling step, greatly reducing computational overhead and inference latency. However, most existing one-step diffusion methods are constrained by the performance of the teacher model, where poor teacher performance results in image artifacts.

To address this limitation, we propose FluxSR, a novel one-step diffusion Real-ISR technique based on flow matching models. We use the state-of-the-art diffusion model FLUX.1-dev as both the teacher model and the base model. First, we introduce Flow Trajectory Distillation (FTD) to distill a multi-step flow matching model into a one-step Real-ISR. Second, to improve image realism and address high-frequency artifact issues in generated images, we propose TV-LPIPS as a perceptual loss and introduce Attention Diversification Loss (ADL) as a regularization term to reduce token similarity in transformer, thereby eliminating high-frequency artifacts. Comprehensive experiments demonstrate that our method outperforms existing one-step diffusion-based Real-ISR methods. The code and model will be released at https://github.com/JianzeLi-114/FluxSR.

---

[*]Equal contribution  [1]Shanghai Jiao Tong University [2]Harvard University [3]Huawei Consumer Business Group [4]Huawei Noah's Ark Lab. Correspondence to: Yulun Zhang[†] <yulun100@gmail.com>.

*Proceedings of the 42nd International Conference on Machine Learning*, Vancouver, Canada. PMLR 267, 2025. Copyright 2025 by the author(s).

# 1. Introduction

Real-world Image Super-Resolution (Real-ISR) (Wang et al., 2020; 2021) aims to recover high-quality images from low-quality ones captured in real-world settings. Traditional image super-resolution (Kim et al., 2016; Zhang et al., 2015; Dong et al., 2016a;b; Chen et al., 2023) assumes a known degradation process. However, this assumption does not account for the complex and unknown degradations present in real-world low-quality images (Wang et al., 2021). Consequently, real-world super-resolution tasks are more challenging and practical. In recent years, they have attracted increasing attention from researchers.

Diffusion models (Ho et al., 2020; Song et al., 2020) are a type of generative model and initially designed for text-to-image (T2I) tasks. They have shown overwhelming advantages in many computer vision tasks (Rombach et al., 2022a). In recent years, numerous researchers have applied diffusion models to Real-ISR (Wang et al., 2024a; Lin et al., 2024; Yang et al., 2024; Yu et al., 2024). These applications have achieved unprecedented quality. These methods leverage the strong priors of pre-trained diffusion models, making the generated images exhibit more realistic details. Very recently, a lot of efforts have been made to investigate the scaling law of diffusion models (Henighan et al., 2020; Yu et al., 2024; Tian et al., 2024) for image generation. Interestingly, a large model, e.g., Flux (Labs, 2023) with 12B parameters, is able to significantly improve the visual quality and photo-realism, compared to those small diffusion models (Rombach et al., 2022b; Podell et al., 2023; Esser et al., 2024) with 1B~3B parameters. Nevertheless, such a large model still requires multiple steps for inference and becomes very computationally expensive, hindering its practical applications. Thus, how to reduce the number of steps to achieve efficient inference based on large diffusion models becomes an important problem.

To address this issue, many one-step distillation methods (Wang et al., 2024b; Wu et al., 2024a; Xie et al., 2024; He et al., 2024; Dong et al., 2024; Zeng et al., 2024) could be useful. But they still suffer from several critical issues, particularly raised by the *generative distribution shift issue* and the training difficulty of *very large model*. **First**, fine-tuning a well-trained T2I model on SR data may easily destroy the original noise-to-image mapping and thus incur a distribution shift, as shown in Figure 2. Note that recent large diffusion models often follow the flow matching strategy (Esser et al., 2024; Labs, 2023) that explicitly learns the flow along the diffusion path. In other words, existing one-step methods may completely ignore the originally well-learned T2I flow when learning the target SR flow. As a result, existing one-step models tend to produce images with unexpected artifacts and degraded visual quality. **Second**, the memory footprint and training cost become

extremely high or even infeasible when distilling a large student model from an additional teacher of at least the same model size. For example, we find that even a server with 8 A800-80GB GPUs cannot satisfy the memory requirement of this distillation if we directly apply the popular one-step distillation method OSEDiff (Wu et al., 2024a) on top of Flux.1-dev (Labs, 2023).

In this paper, we propose a novel one-step diffusion model for Real-ISR, called FluxSR, with FLUX.1-dev as the base model. Specifically, our design comprises three main components: 1) We propose a Flow Trajectory Distillation (FTD) to address the generative distribution issue. The key idea is to build the relationship between the noise-to-image flow in T2I and LR-to-HR flow in SR based on the flow matching theory. Unlike existing methods, we explicitly keep the original T2I flow unchanged while learning the SR flow trajectory conditioned on it. This approach maximizes the preservation of the teacher model's generative capabilities, thereby enhancing the realism of the generated images. 2) We develop a large model friendly training strategy that does not rely on an extra teacher model to compute the distillation loss. Instead, we cast the knowledge of the teacher model into the noise-to-image flow in the T2I task. In this sense, we are able to generate a bunch of flow data in the offline mode and exclude the teacher model from training to save memory consumption. 3) We propose TV-LPIPS as a perceptual loss. By incorporating the idea of total variation (TV), this loss emphasizes the restoration of high-frequency components and reduces artifacts in the generated images. Moreover, we introduce the Attention Diversification Loss (ADL) (Guo et al., 2023) that improves the diversity of different tokens in attention modules. We use it as a regularization term to address the repetitive patterns observed in the images. Extensive experiments show that our FluxSR achieves remarkable performance and requires only one sampling step. Figure 1 presents the visual results of our method. In summary, our contributions are as follows:

- We develop FluxSR, a one-step diffusion Real-ISR model based on FLUX.1-dev. To the best of our knowledge, this is the first one-step diffusion for Real-ISR based on a large model with over 12B parameters.

- We propose a Flow Trajectory Distillation (FTD) method that explicitly builds the relationship between the noise-to-image flow and LR-to-HR flow. With the noise-to-image flow unchanged, we are able to preserve the high photo-realism in the T2I model and effectively transfer it to the LR-to-HR flow for SR.

- To make the training feasible, we propose a large model friendly training strategy that excludes the extra teacher model from the training phase. Instead, we cast the knowledge from teacher into the noise-to-image flow and generate a bunch of them in the offline mode, to reduce both memory consumption and training cost.

## 2. Related Work

### 2.1. Acceleration of Flow Matching Models

Liu et al. (2022) proposed the Rectified Flow method, which straightens the flow trajectory to achieve high-quality results within a one sampling step, laying a solid theoretical foundation for subsequent research. InstaFlow (Liu et al., 2023) applies the Reflow method to straighten the curved ODE solving path, allowing latents to transition more quickly from the noise distribution to the image distribution. The straightened ODE path also reduces the learning difficulty for the student model, improving the distillation effectiveness. This enables one-step generation for large-scale text-to-image tasks. PeRFlow (Yan et al., 2024) further improves Reflow correction by segmenting the flow trajectory, achieving exceptional performance.

### 2.2. Diffusion-based Real-ISR

**Multi-step Diffusion-based Real-ISR.** In recent years, diffusion models have achieved remarkable success in the field of image super-resolution (Wang et al., 2024a; Lin et al., 2024; Yang et al., 2024; Yue et al., 2024; Wu et al., 2024b; Yu et al., 2024). DiffBIR (Lin et al., 2024) reconstructs low-resolution (LR) images using a small network and then employs ControlNet (Zhang et al., 2023) to control the generation of the diffusion model. SeeSR (Wu et al., 2024b) introduces a module for extracting semantic information from images. This module effectively guides the diffusion model's generation through semantic cues, preventing errors caused by image degradation. SUPIR (Yu et al., 2024) uses Restoration-Guided Sampling to ensure both generative capability and fidelity. It also leverages a large dataset and a large pre-trained diffusion model, SDXL (Podell et al., 2023), to enhance the model's performance.

**One-step Diffusion-based Real-ISR.** Recently, one-step diffusion ISR models have become a popular research direction, showing great potential and application value (Wang et al., 2024b; Wu et al., 2024a; Xie et al., 2024; He et al., 2024; Dong et al., 2024). SinSR (Wang et al., 2024b) introduces a deterministic sampling method. It fixes the noise-image pair using consistency-preserving distillation. OSEDiff (Wu et al., 2024a) employs Variational Score Distillation (VSD) (Wang et al., 2024c; Nguyen & Tran, 2024) and directly uses the low-resolution (LR) image as the starting point for diffusion inversion. In addition, OSEDiff uses DAPE (Wu et al., 2024b) to extract semantic information from the LR image as the generation condition. ADDSR (Xie et al., 2024) combines adversarial training by introducing Adversarial Diffusion Distillation (ADD) and ControlNet to achieve both 4-step and one-step models. TSD-SR (Dong et al., 2024) proposes Target Score Distillation (TSD) and a Distribution-Aware Sampling Module (DASM), effectively addressing the issue of artifacts caused by VSD in the early stages of training.

## 3. Background

### 3.1. Flow Matching Models

Given two data distributions $p_0$ and $p_1$, there exists a vector field $u_t$ that generates a probabilistic path $p_t$ transitioning from $p_0$ to $p_1$. In generative models, $p_0$ represents the data distribution, while $p_1$ is an easily accessible simple distribution, such as the standard normal distribution $\mathcal{N}(0,1)$.

Following Esser et al. (2024), we define the forward process as:

$$x_t = a_t x_0 + b_t \epsilon, \quad \text{where } \epsilon \sim \mathcal{N}(0,1). \tag{1}$$

The coefficients $a_t$ and $b_t$ satisfy $a_0 = 1$, $b_0 = 0$, $a_1 = 0$, and $b_1 = 1$. This defines a probabilistic path $p_t$ from $p_0$ to $p_1$. The transformed variable is given by:

$$x_t' = u_t(x_t|\epsilon) = \frac{a_t'}{a_t} x_t - \epsilon b_t \left( \frac{a_t'}{a_t} - \frac{b_t'}{b_t} \right). \tag{2}$$

Subsequently, the marginal vector field $u_t(x_t)$ is obtained using the conditional vector field $u_t(x_t|\epsilon)$ as follows:

$$u_t(x_t) = \int u_t(x_t|\epsilon) \frac{p(x_t|\epsilon)p(\epsilon)}{p_t(x_t)} \, d\epsilon. \tag{3}$$

Here, the marginal probability density $p_t(x_t)$ is defined by:

$$p_t(x_t) = \int p_t(x_t|\epsilon)p(\epsilon) \, d\epsilon. \tag{4}$$

Flow matching aims to train a vector field $v_\theta(x,t)$, parameterized by a deep neural network, to approximate the marginal vector field $u_t(x_t)$. Specifically, flow matching minimizes the following objective (Lipman et al., 2022):

$$\mathcal{L}_{\text{FM}}(\theta) := \mathbb{E}_{t, \, p_t(x_t)} \| v_\theta(x_t, t) - u_t(x_t) \|^2. \tag{5}$$

However, the expression for $u_t$ cannot be explicitly computed, making the direct optimization of the aforementioned loss challenging. Lipman et al. (2022) proposed conditional flow matching, demonstrating that we can optimize the following equivalent yet more tractable objective by using $u_t(x_t|\epsilon)$:

$$\mathcal{L}_{\text{CFM}}(\theta) := \mathbb{E}_{t, \, p_t(x_t|\epsilon), \, p(\epsilon)} \| v_\theta(x_t, t) - u_t(x_t|\epsilon) \|^2. \tag{6}$$

### 3.2. Flow Trajectories

In this paper, we consider the flow trajectory used in FLUX.1-dev, namely rectified flow (ReFlow) (Liu et al., 2022). This is a simple diffusion trajectory that defines the forward process as a straight path between the data distribution and the noise distribution (Liu et al., 2022; Esser et al., 2024), specifically:

$$x_t = (1-t)x_0 + t\epsilon, \tag{7}$$

where $x_0 \sim p_0$, $\epsilon \sim p_1 = \mathcal{N}(0,1)$.

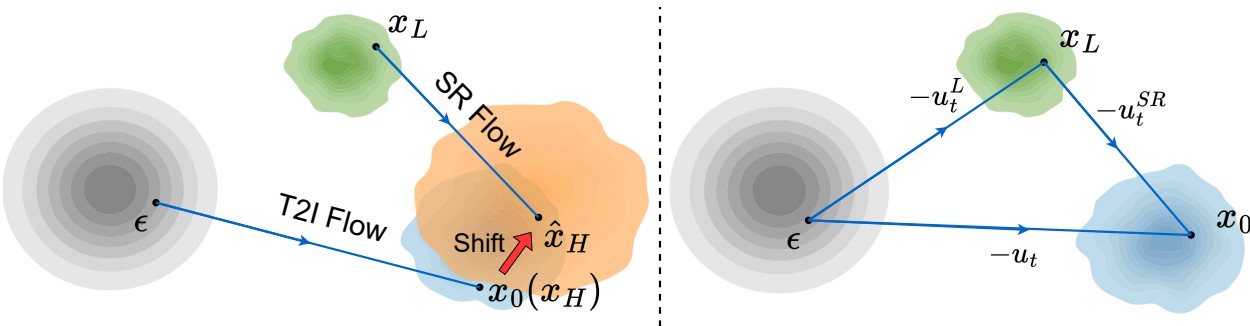

Figure 2: Difference of exiting methods and our Flow Trajectory Distillation. (Left) Based on the pre-trained models from noise $\epsilon$ to images $x_0$, existing one-step diffusion models fine-tune the model from LR images to HR images $x_H$. It may lead to a distribution shift between the real data distribution (blue) and the generated distribution (orange). (Right) To bridge the mapping from LR image distribution (green) to real data distribution, we propose Flow Trajectory Distillation. We constrain $u_t^{SR}$ using the other two trajectories in the triangle, ensuring that the real data distribution (blue) does not shift.

By substituting into Equation 2, we obtain the conditional vector field of ReFlow:

$$u_t(x_t|\epsilon) = \frac{\epsilon - x_t}{1 - t} = \epsilon - x_0. \tag{8}$$

Therefore, following (Lipman et al., 2022; Esser et al., 2024), the training objective of ReFlow is

$$\mathcal{L}_{\text{ReFlow}}(\theta) = \mathbb{E}_{t, p_t(x_t|\epsilon), p(\epsilon)} \|v_\theta(x_t, t) - (\epsilon - x_0)\|_2^2. \tag{9}$$

Intuitively, the goal of ReFlow is to train the neural network $v_\theta(x_t, t)$ to predict the velocity from noise to data samples.

## 4. Method

### 4.1. Flow Trajectory Distillation (FTD)

Our goal is to distill a one-step diffusion super-resolution model from a pre-trained text-to-image (T2I) flow model. Most current one-step diffusion ISR methods directly fine-tune the pre-trained T2I model and incorporate modules such as VSD or GANs to improve performance (Wu et al., 2024a; Xie et al., 2024; Dong et al., 2024). Although these methods have achieved good results, they still face some challenges. As shown on the left side of Figure 2, the flow trajectory of the pre-trained T2I model is not aligned with that of the SR model. During fine-tuning, these methods have no mechanism to keep the diffusion endpoint distribution unchanged. In other words, the real data distribution (blue) in the figure shifts, converting to the generated distribution (orange). For large-scale T2I models, which have already fit the real data distribution well, fine-tuning them using the above methods could lead to negative outcomes.

Ideally, the resulting model serves as a mapping from the low-resolution (LR) image distribution $p_L$ (green distrubition in Figure 2) to the high-resolution (HR) image distribution $p_0$ (blue distrubition in Figure 2). We aim to fix

the distribution of the vector field $u_t^{SR}$ at $x_0$ while modifying the distribution of the diffusion starting point (i.e., transitioning from the noise distribution to the LR image distribution as shown in Figure 2) by fine-tuning the T2I model. Therefore, we propose Flow Trajectory Distillation, which indirectly obtains $u_t^{SR}$ by fitting $u_t^L$, avoiding the shift in the real data distribution.

**Approximating the LR Image Distribution.** Inspired by DMD (Yin et al., 2024b;a), we can learn the underlying distribution of the training data by training a diffusion model. For flow matching models, training on LR data allows us to obtain parameters $v_\phi$, which fit the vector field $u_t^L$ that maps the noise distribution to the LR image distribution. The corresponding conditional flow trajectory is given by:

$$x_t = (1 - t)x_L + t\epsilon, \tag{10}$$

where $x_L \sim p_L$, $\epsilon \sim \mathcal{N}(0,1)$ and $t \in [0,1]$. The velocity of a sample $x_t$ at time $t$ is given by $v_\phi(x_t, t)$.

**Computing the LR-to-HR Flow from Noise-to-Image Flow.** At this point, we have obtained the flow model $v_\phi$ that maps from noise to low-resolution (LR) images and the flow model $v_{\text{real}}$ that maps from noise to real-world high-resolution (HR) images (the pre-trained T2I model). Given the linearity of the ReFlow trajectory, we can easily derive the flow model $v_\theta$ for mapping LR images to HR images. We have:

$$x_0 = \epsilon - u_t, \quad x_L = \epsilon - u_t^L. \tag{11}$$

Here, $v_{\text{real}}$ and $v_\phi$ parameterize $u_t$ and $u_t^L$, respectively. By combining the above equations, we obtain the trajectory from $x_L$ to $x_0$:

$$x_0 = x_L - (u_t - u_t^L). \tag{12}$$

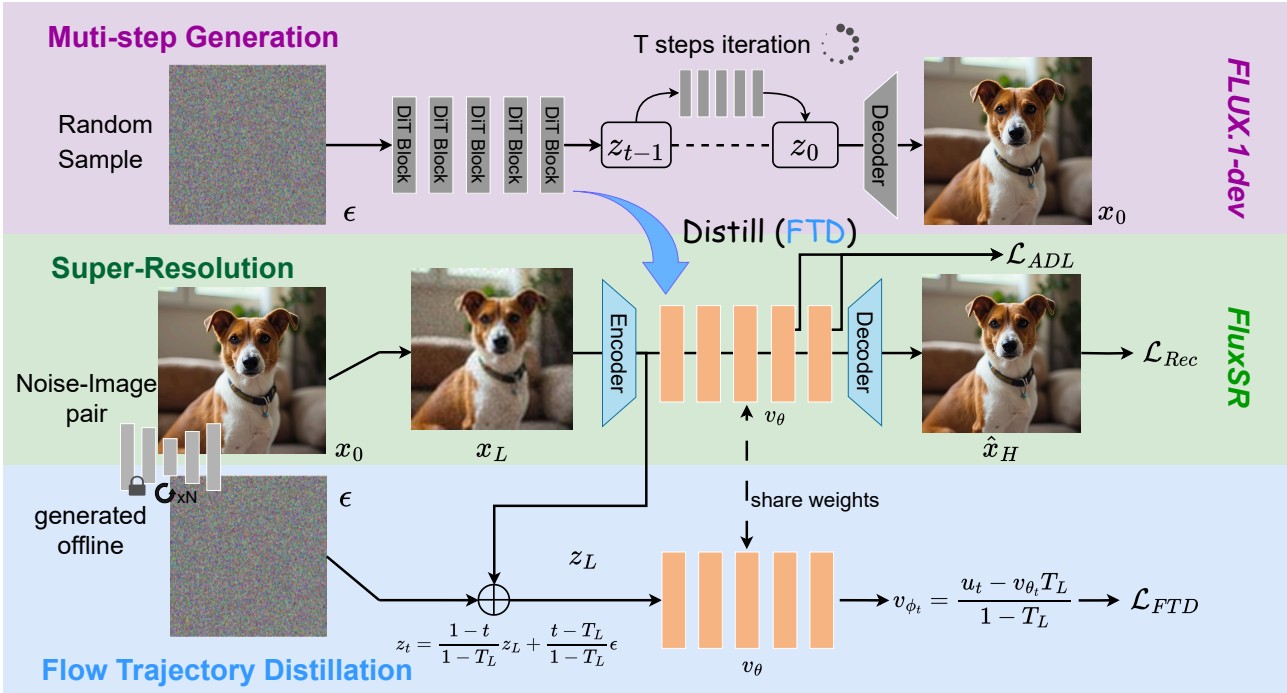

Figure 3: Training framework of FluxSR. (Top) Multi-step inference process of the pre-trained FLUX model. (Middle) Training strategy of FluxSR. (Bottom) Computation process of FTD. We distill a one-step super-resolution model from the multi-step FLUX model, without the need for the teacher model to be involved online during training.

### 4.2. Large Model Friendly Training Strategy

Although we have derived the theoretical formulation of FTD, its practical application faces the following challenges: *i)* **Inference Efficiency:** During inference, we need both the vector field $u_t$ calculated by the pre-trained T2I model and the vector field $u_t^L$ calculated by the model fine-tuned on LR data. This requires two different flow models with separate parameters, leading to significant computational overhead during inference. *ii)* **Estimation Error:** Running the flow model in a one step makes it difficult to accurately estimate the velocity at time $t$. Without using a reconstruction loss to optimize the generator, the model performance may degrade. In this section, we propose an optimized training strategy to ensure that only a one flow model is required during inference. Additionally, we incorporate a reconstruction loss to enhance model performance.

**Direct Parameterization of $u_t^{SR}$.** As shown on the left side of Figure 3, since we can derive $u_t^{SR}$ from $u_t$ and $u_t^L$, we can also obtain $u_t^L$ from $u_t$ and $u_t^{SR}$. This avoids the issue caused by the inability to directly parameterize $u_t^{SR}$. We parameterize $u_t^{SR}$ using $v_\theta$. To represent both $u_t^L$ and $u_t^{SR}$ with a one model, we define the time step corresponding to the LR image as $T_L$ instead of 0. This ensures that the model represents only $u_t^L$ in the time range $[T_L, 1]$ and only $u_t^{SR}$ at $T_L$. Additionally, the LR image distribution is more similar to the intermediate states $x_t$ of the pre-trained diffusion

model. As shown in Figure 3, similar to Eq. (11), we have:

$$\begin{cases} x_0 = \epsilon - u_t, \\ x_L = \epsilon - (1 - T_L)u_t^L, & (13) \\ x_L - x_0 = u_t^{SR}T_L. \end{cases}$$

By combining the above equations, we obtain:

$$u_t^L = \frac{u_t - u_t^{SR}T_L}{1 - T_L}, \quad \text{where } t \in [T_L, 1]. \quad (14)$$

The model parameterization can be expressed as:

$$v_{\phi_t}(x_t, t) = \frac{u_t(x_t|\epsilon) - v_{\theta_t}(x_t, t)T_L}{1 - T_L}, \quad (15)$$

where

$$x_t = \frac{1-t}{1-T_L}x_L + \frac{t-T_L}{1-T_L}\epsilon, \quad t \in [T_L, 1]. \quad (16)$$

**Generating noise-to-image flow for distillation.** We pre-compute noise-sample pairs generated by FLUX and use them as training data, without relying on any real images. This approach offers two crucial benefits for large model training. 1) By using data pairs generated by the teacher model, we can directly compute $u_t(x_t|\epsilon) = \epsilon - x_0$, thus avoiding the estimation error during single-step inference. 2) The teacher model is not required for online inference during training, which significantly reduces GPU usage and

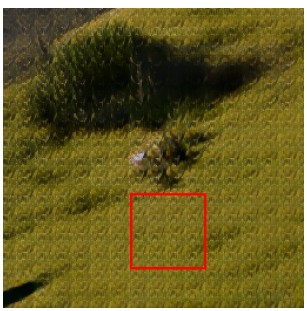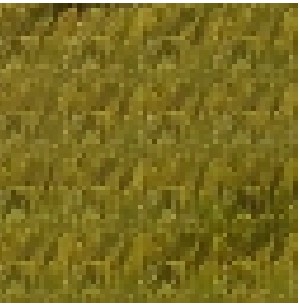

Figure 4: Examples of Pronounced Periodic Artifacts During Training. Left: 256-pixel image with noticeable periodic high-frequency artifacts. Right: 64-pixel zoomed-in region, showing artifacts with four cycles in both width and height.

training time, especially for large T2I models like FLUX. Using $v$-prediction, the loss function of FTD is given by:

$$\mathcal{L}_{\text{FTD}}(\theta) = \mathbb{E}_{t,\,p_t(x_t|\epsilon),\,p(\epsilon)} \left\| (1 - T_L)v_{\phi_t} - (\epsilon - x_L) \right\|_2^2$$
$$= \mathbb{E}_{t,\,p_t(x_t|\epsilon),\,p(\epsilon)} \left\| (u_t - v_\theta(x_t, t)T_L) - (\epsilon - x_L) \right\|_2^2 \quad (17)$$

where $u_t = \epsilon - x_0$, $t \in [T_L, 1]$.

The generator $G_\theta$ can be expressed as:

$$G_\theta(x_L) = x_L - v_\theta(x_L, T_L)T_L. \quad (18)$$

### 4.3. Anti-artifacts Loss Functions.

During training, we observe that the generator's predictions exhibit periodic high-frequency artifacts in the pixel space. As shown in Figure 4, the artifact period is 16 pixels, exactly the product of the VAE scaling factor (8) and the transformer patch size (2). This indicates that each token has similar components in certain dimensions.

**Improvement of Perceptual Loss.** We aim to reduce variations between adjacent pixels in flat regions to suppress high-frequency artifacts while preserving sharp edges. Inspired by the total variation (TV) loss, we propose TV-LPIPS as the perceptual loss for training. Specifically, TV-LPIPS is computed as follows:

$$\mathcal{L}_{\text{TV}_{LPIPS}}(I, I_0) = \mathcal{L}_{\text{LPIPS}}(I, I_0) + \gamma \mathcal{L}_{\text{LPIPS}}(TV(I), TV(I_0)), \quad (19)$$

where

$$TV(I_{i,j}) = (|I_{i+1,j} - I_{i,j}| + |I_{i,j+1} - I_{i,j}|). \quad (20)$$

TV-LPIPS measures the degree of pixel variation and computes the LPIPS distance with the ground-truth. This not only prevents excessive variations between adjacent pixels in smooth regions but also enhances the LPIPS loss's sensitivity to high-frequency components. In summary, the reconstruction loss for training is given by:

$$\mathcal{L}_{\text{Rec}}(G_\theta(x_L), x_H) = \mathcal{L}_{\text{MSE}}(G_\theta(x_L), x_H) + \lambda \mathcal{L}_{\text{TV}_{LPIPS}}(G_\theta(x_L), x_H). \quad (21)$$

**Attention Diversification Loss.** To address periodic artifacts at the feature level, we introduce the Attention Diversification Loss (ADL) proposed by Guo et al. (2023). ADL aims to reduce similarity between tokens and enhance attention diversity. We incorporate this loss to prevent different tokens from generating identical feature components.

To reduce computational complexity, ADL first approximates the overall cosine similarity by computing the cosine similarity between each token feature vector $A_i^{(l)}$ and the mean of all token feature vectors, defined as:

$$\bar{A}^{(l)} = \frac{1}{N} \sum_{i=1}^{N} A_i^{(l)}. \quad (22)$$

Here, $A_i^{(l)}$ represents the $i$-th feature vector in the output of the $l$-th transformer layer. For a model with $L$ layers, ADL computes the mean ADL loss across all layers:

$$\mathcal{L}_{\text{ADL}} = \frac{1}{L} \sum_{l=1}^{L} \mathcal{L}_{\text{ADL}}^{(l)}, \quad \mathcal{L}_{\text{ADL}}^{(l)} = \frac{1}{N} \sum_{i=1}^{N} \frac{A_i^{(l)} \cdot \bar{A}^{(l)}}{\|A_i^{(l)}\| \|\bar{A}^{(l)}\|}. \quad (23)$$

In summary, the overall training procedure of FluxSR is presented in Algorithm 1.

---

**Algorithm 1** FluxSR Training Procedure

---

1: **Input:** Pre-computed noise-image dataset $\mathcal{D} = \{\epsilon, x_0, z_0\}$. Pre-trained diffusion model $v_\psi$ and VAE encoder $E_\psi$, decoder $D_\psi$. Training iterations $N$.
2: **Output:** one-step generator $G_\theta$.
3: **Init:** $v_\theta \leftarrow v_\psi$, $E_\theta \leftarrow E_\psi$, $D_\theta \leftarrow D_\psi$. Initialize: Trainable LoRA mounted on $v_\theta$.
4: **for** $i = 1$ to $N$ **do**
5:     Sample $(\epsilon, x_0, z_0) \sim \mathcal{D}$.
6:     $u_t \leftarrow \epsilon - z_0$
7:     // FTD Loss:
8:     Sample $t \in [T_L, 1]$.
9:     $x_t \leftarrow \dfrac{1-t}{1-T_L}x_L + \dfrac{t-T_L}{1-T_L}\epsilon$
10:     $v_{\phi_t}(z_t, t) \leftarrow \dfrac{u_t - v_{\theta_t}(z_t, t)T_L}{1-T_L}$
11:     Compute $\mathcal{L}_{\text{FTD}}$ using Eq. (17).
12:     // Reconstruction Loss:
13:     $\hat{z}_0 \leftarrow z_L - (v_\theta(z_L, T_L))T_L$.
14:     $\hat{x}_0 \leftarrow D_\theta(\hat{z}_0)$.
15:     Compute $\mathcal{L}_{\text{TV}_{LPIPS}}$ using Eq. (19)
16:     $\mathcal{L}_{\text{Rec}} = \mathcal{L}_{\text{MSE}} + \lambda \mathcal{L}_{\text{TV}_{LPIPS}}$.
17:     // ADL Loss:
18:     Compute $\mathcal{L}_{\text{ADL}}$ using Eq. (23).
19:     $\mathcal{L}(\theta) = \mathcal{L}_{\text{FTD}} + \mathcal{L}_{\text{Rec}} + \mu \mathcal{L}_{\text{ADL}}$
20:     Update $v_\theta$ using $\mathcal{L}(\theta)$.
21: **end for**

---

Table 1: Quantitative results (×4) on the Real-ISR testset with ground truth. The best and second-best results are colored red and blue. In the one-step diffusion models, the best metric is **bolded**.

| Model | Method | PSNR↑ | SSIM↑ | LPIPS↓ | DISTS↓ | MUSIQ↑ | MANIQA↑ | TOPIQ↑ | Q-Align↑ |
|---|---|---|---|---|---|---|---|---|---|
| **RealSR** | StableSR-s200 | 26.28 | 0.7733 | 0.2622 | 0.1583 | 60.53 | 0.3706 | 0.5036 | 3.8789 |
| | DiffBIR-s50 | 24.87 | 0.6486 | 0.3834 | 0.2015 | 68.02 | 0.5287 | 0.6618 | 4.1244 |
| | SeeSR-s50 | 26.20 | 0.7555 | 0.2806 | 0.1784 | 66.37 | 0.5089 | 0.6565 | 3.9862 |
| | ResShift-s15 | 25.45 | 0.7246 | 0.3727 | 0.2344 | 56.18 | 0.3477 | 0.4420 | 3.8936 |
| | ADDSR-s4 | 23.15 | 0.6662 | 0.3769 | 0.2353 | 66.54 | 0.6094 | 0.7241 | 4.1635 |
| | SinSR-s1 | **25.83** | 0.7183 | 0.3641 | 0.2193 | 61.62 | 0.4255 | 0.5362 | 3.9237 |
| | OSEDiff-s1 | 24.57 | 0.7202 | 0.3036 | **0.1808** | 67.31 | 0.4775 | 0.6382 | 4.0646 |
| | ADDSR-s1 | 25.23 | **0.7295** | 0.2990 | 0.1852 | 63.08 | 0.4093 | 0.5685 | 3.9806 |
| | TSD-SR-s1 | 23.80 | 0.6987 | **0.2874** | 0.1843 | 68.31 | 0.4899 | 0.6568 | 4.0926 |
| | **FluxSR-s1** | 24.83 | 0.7175 | 0.3200 | 0.1910 | **68.95** | **0.5335** | **0.6699** | **4.3781** |
| **DIV2K-val** | StableSR-s200 | 23.68 | 0.6270 | 0.4167 | 0.2023 | 49.51 | 0.2696 | 0.3765 | 3.7427 |
| | DiffBIR-s50 | 22.33 | 0.5133 | 0.4681 | 0.1889 | 70.07 | 0.5471 | 0.6958 | 4.2666 |
| | SeeSR-s50 | 23.21 | 0.6114 | 0.3477 | 0.1706 | 67.99 | 0.4687 | 0.6592 | 4.4594 |
| | ResShift-s15 | 23.55 | 0.6023 | 0.4088 | 0.2228 | 56.07 | 0.3409 | 0.4580 | 3.9961 |
| | ADDSR-s4 | 22.08 | 0.5578 | 0.4169 | 0.2145 | 68.26 | 0.5496 | 0.7168 | 4.3910 |
| | SinSR-s1 | 22.55 | 0.5405 | 0.4390 | 0.2033 | 62.25 | 0.4241 | 0.5787 | 4.1712 |
| | OSEDiff-s1 | **23.10** | 0.6127 | 0.3447 | 0.1750 | 66.62 | 0.4115 | 0.5971 | 4.1366 |
| | ADDSR-s1 | 22.74 | 0.6007 | 0.3961 | 0.1974 | 62.08 | 0.3867 | 0.5817 | 4.2971 |
| | TSD-SR-s1 | 21.65 | 0.5546 | 0.3456 | **0.1530** | 68.65 | 0.4393 | 0.6415 | 4.1539 |
| | **FluxSR-s1** | 22.30 | **0.6177** | **0.3397** | 0.1634 | **68.72** | **0.4615** | **0.6426** | **4.6128** |

Table 2: Quantitative results (×4) on RealSet65 testset. The best and second-best results are colored red and blue. In the one-step diffusion models, the best metric is **bolded**.

| Method | MUSIQ↑ | MANIQA↑ | TOPIQ↑ | Q-Align↑ |
|---|---|---|---|---|
| StableSR-s200 | 58.89 | 0.3535 | 0.4974 | 3.8093 |
| DiffBIR-s50 | 71.23 | 0.5682 | 0.7015 | 4.1599 |
| SeeSR-s50 | 69.79 | 0.5030 | 0.6774 | 4.1172 |
| ResShift-s15 | 59.36 | 0.3622 | 0.4953 | 3.8942 |
| ADDSR-s4 | 68.97 | 0.5613 | 0.6971 | 4.1672 |
| SinSR-s1 | 64.22 | 0.4462 | 0.5947 | 4.0390 |
| OSEDiff-s1 | 69.04 | 0.4625 | 0.5969 | 4.1065 |
| ADDSR-s1 | 64.22 | 0.3947 | 0.5616 | 4.0806 |
| TSD-SR-s1 | 69.34 | 0.4893 | 0.6392 | 3.9936 |
| **FluxSR-s1** | 70.75 | **0.5495** | **0.6670** | **4.2134** |

## 5. Experiments

### 5.1. Experimental Settings

**Training Datasets.** Our method does not require any real datasets. We generate 2400 noise-image pairs of size 1024x1024 using FLUX.1-dev (Labs, 2023) as training data. To obtain the corresponding low-resolution (LR) images, we use the degradation pipeline proposed by RealESR-GAN (Wang et al., 2021).

**Test Datasets.** We evaluate our model on the synthetic dataset DIV2K-val (Agustsson & Timofte, 2017) and two real datasets: RealSR (Cai et al., 2019) and RealSet65 (Yue et al., 2024). From DIV2K-val, we use the RealESRGAN degradation pipeline to generate corresponding LR images. On the these datasets, we evaluate using full-size images to assess the model's performance in real-world scenarios.

**Compared Methods and Metrics.** We compare the performance of our model with other diffusion-based ISR models, including multi-step diffusion ISR models: StableSR (Wang et al., 2024a), DiffBIR (Lin et al., 2024), SeeSR (Wu et al., 2024b), ResShift (Yue et al., 2024), and AddSR (Xie et al., 2024); and one-step diffusion ISR models: SinSR (Wang et al., 2024b), OSEDiff (Wu et al., 2024a), and TSD-SR (Dong et al., 2024). We evaluate our model and the aforementioned methods using 4 full-reference metrics: PSNR, SSIM, LIPIS (Zhang et al., 2018), and DISTS (Ding et al., 2020), as well as 4 no-reference metrics: MUSIQ (Ke et al., 2021), MANIQA (Yang et al., 2022), TOPIQ (Chen et al., 2024), and Q-Align (Wu et al., 2023). PSNR and SSIM are computed on the Y channel in the YCbCr space.

### 5.2. Comparison with State-of-the-Art Methods

**Quantitative Comparisons.** Tables 1 and 2 presents a quantitative comparison between FluxSR and other diffusion-based Real-ISR methods. Among one-step methods, our approach achieves the best performance across all no-reference (NR) metrics on all test datasets. For FR metrics like PSNR and SSIM, recent studies have demonstrated that image fidelity and perceptual quality involve a trade-off. In the context of diffusion-based super-resolution methods, PSNR and SSIM have limited reference value. Compared to multi-step methods, FluxSR outperforms StableSR across all datasets. Against DiffBIR, SeeSR, and AddSR, FluxSR shows slightly lower performance in TOPIQ. Additionally, we provide further comparisons with non-diffusion-based methods in the supplementary material.

**Qualitative Comparisons.** Figure 5 presents visual comparison between FluxSR and other methods. FluxSR is capable of generating realistic details under severe degradation.

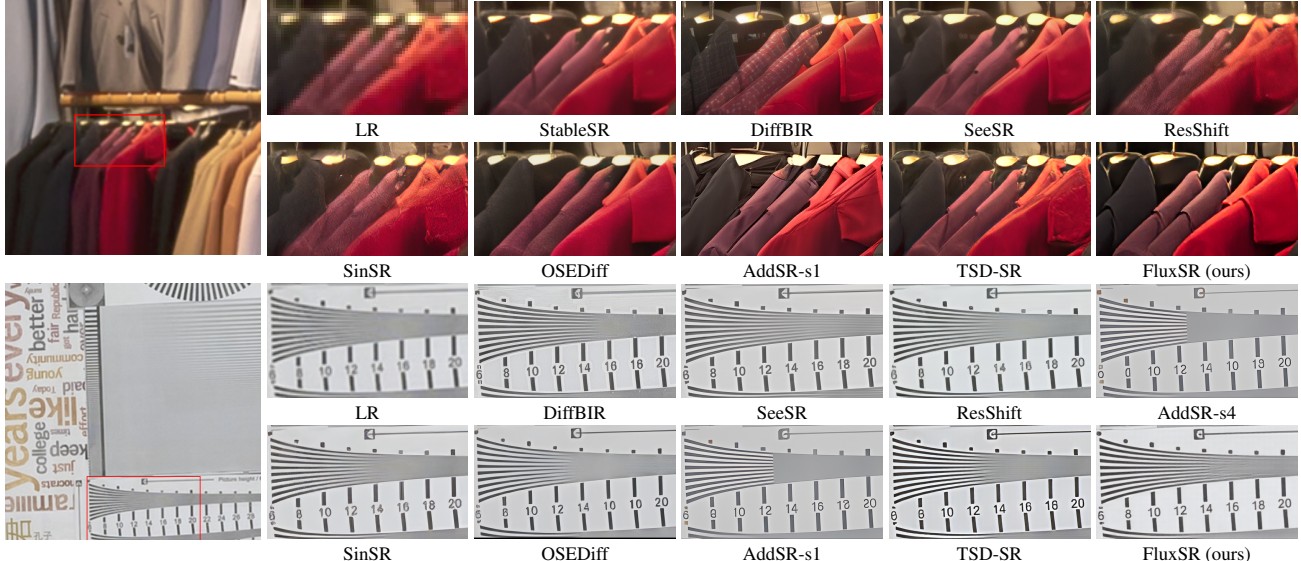

Figure 5: Visual comparisons (×4) on Real-ISR task.

Table 3: Ablation study on FTD.

| Method | PSNR↑ | MUSIQ↑ | MANIQA↑ | Q-Align↑ |
|---|---|---|---|---|
| w/o FTD | **26.33** | 56.02 | 0.3775 | 3.5170 |
| FTD (ours) | 24.67 | **67.84** | **0.5203** | **4.1473** |

Table 4: Ablation study on different loss functions.

| $\mathcal{L}_{LPIPS}$ | $\mathcal{L}_{TV\text{-}LPIPS}$ | $\mathcal{L}_{EA\text{-}DISTS}$ | $\mathcal{L}_{ADL}$ | PSNR↑ | MUSIQ↑ | MANIQA↑ | Q-Align↑ |
|---|---|---|---|---|---|---|---|
| ✓ | | | | 23.10 | 64.55 | 0.4937 | 4.0515 |
| | ✓ | | | 22.09 | 65.04 | 0.5113 | 4.0927 |
| | | ✓ | | 23.67 | 64.83 | 0.5036 | 4.0003 |
| | | ✓ | ✓ | **24.72** | 67.13 | 0.5138 | 4.0691 |
| | ✓ | | ✓ | 24.67 | **67.84** | **0.5203** | **4.1473** |

For example, in the first row of Figure 5, which depicts the restoration of a coat image, DiffBIR, ResShift, and SinSR are affected by noise, resulting in artificial textures. Although AddSR and TSD-SR generate relatively sharp images, they fail to accurately restore the collar's design. In contrast, FluxSR reconstructs the collar in a way that closely resembles the real-world appearance. The second row of Figure 5 demonstrates the restoration of numerical digits. FluxSR produces the most realistic result. While TSD-SR also approximately restores the digits, it suffers from Sinc noise, generating bright edges around the numbers.

### 5.3. Ablation Study

In this section, we use RealSR as the test dataset. The training iterations are set to 30k. Other settings remain consistent with those mentioned in Sec. 5.1.

**Effectiveness of FTD loss.** To verify the effectiveness and of FTD, we compare it with training using only the reconstruction loss, as shown in Table 3. Training the one-step flow model with only the reconstruction loss results in

poor performance, failing to generate high-frequency details and exhibiting significant high-frequency artifacts. Using the proposed FTD loss does not disrupt the data distribution learned by the teacher model. It effectively restores high-frequency details and achieves a higher degree of realism.

**Effectiveness of ADL and TV-LPIPS.** To verify the effectiveness of ADL and the proposed TV-LPIPS loss, we conducted relevant ablation experiments to investigate the impact of each loss function component. We also included the use of EA-DISTS, proposed by DFOSD, as a perceptual loss. Table 4 presents the experimental results, showing that using TV-LPIPS as a perceptual loss and ADL as a regularization term achieves the best performance.

## 6. Conclusion and Limitation

This paper proposes FluxSR, an efficient one-step Real-ISR model based on FLUX, the state-of-the-art T2I diffusion model. FluxSR leverages Flow Trajectory Distillation (FTD) to distill a multi-step flow matching model into a one-step super-resolution model. It is trained using noise-image pairs generated by a fixed multi-step model and does not require any real data. We employ TV-LPIPS and ADL to enhance high-frequency components in the generated images and reduce periodic artifacts. Our experiments demonstrate that FluxSR achieves unprecedented realism.

**Limitations.** Although FluxSR achieves strong performance, it has a large number of parameters and high computational cost. Moreover, we have not entirely eliminated the periodic artifacts mentioned in Section 4.3. In the future, we plan to apply model pruning techniques to compress the model and develop more effective algorithms to prevent periodic artifacts, aiming to achieve a lightweight yet high-performance Real-ISR model.

## Impact Statement

This paper presents work whose goal is to advance the field of Machine Learning. There are many potential societal consequences of our work, none which we feel must be specifically highlighted here.

## Acknowledgments

This work was supported by Shanghai Municipal Science and Technology Major Project (2021SHZDZX0102) and the Fundamental Research Funds for the Central Universities.

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
