# OpenReview forum: "One Diffusion Step to Real-World Super-Resolution via Flow Trajectory Distillation"
_ICML.cc/2025/Conference — ICML 2025 poster_

### Official Review · Reviewer_mAgS · 2025-03-11

**Overall Recommendation:** 3

**Summary:**

The paper introduces FluxSR, a novel one-step diffusion model for real-world image super-resolution (ISR), leveraging flow trajectory distillation (FTD) to distill a multi-step diffusion model into a one-step model. The authors propose several innovations, including TV-LPIPS as a perceptual loss and attention diversification loss (ADL) to reduce high-frequency artifacts. The method achieves promising performance in both quantitative and qualitative evaluations, outperforming existing one-step and multi-step diffusion-based Real-ISR methods.

**Claims And Evidence:**

The paper provides ample empirical evidence to support its claims, including quantitative results, qualitative comparisons, ablation studies, and theoretical justifications. The proposed method, FluxSR, demonstrates its improvements over existing approaches in terms of image quality, computational efficiency, and artifact reduction. However, the paper also acknowledges some limitations, such as the computational cost of training and the presence of periodic artifacts, which could be addressed in future work.

**Essential References Not Discussed:**

The references are relatively comprehensive.

**Experimental Designs Or Analyses:**

The paper employs a comprehensive set of experimental designs and analyses to validate the effectiveness of the proposed FluxSR model. The experiments include quantitative evaluations, qualitative comparisons, ablation studies, and comparisons with non-diffusion methods. The results demonstrate that FluxSR achieves SOTA performance in real-world ISR with only one diffusion step, while also addressing key challenges such as computational efficiency and artifact reduction. The ablation studies provide insights into the contributions of individual components (FTD, TV-LPIPS, ADL), and the visual comparisons highlight the model's ability to generate realistic and detailed images.

**Methods And Evaluation Criteria:**

The paper employs a comprehensive set of methods and evaluation criteria to demonstrate the effectiveness of FluxSR.

- The proposed FTD, TV-LPIPS perceptual loss, and ADL are key innovations that contribute to the model's superior performance.

- The evaluation includes both quantitative metrics and qualitative comparisons, along with thorough ablation studies to validate the contributions of each component.

- The results show that FluxSR achieves SOTA performance in real-world ISR with only one diffusion step.

**Other Comments Or Suggestions:**

None.

**Other Strengths And Weaknesses:**

- Strengths

1) The proposed flow trajectory distillation (FTD) is a novel approach that effectively bridges the gap between noise-to-image and LR-to-HR flows, preserving the generative capabilities of the teacher model while enabling efficient one-step inference.

2) The method achieves impressive results, outperforming existing one-step and multi-step diffusion-based methods across multiple datasets. The qualitative results demonstrate that FluxSR generates more realistic and detailed images compared to other SOTA methods.

3) By reducing the inference steps to one, FluxSR significantly reduces computational overhead and inference latency, making it more practical for real-world applications.

- Weaknesses

1) While the method reduces inference steps, the training process still requires significant computational resources, particularly due to the use of large models like FLUX.1-dev. This could limit its accessibility for researchers with limited resources.

2) Although the authors propose ADL and TV-LPIPS to address high-frequency artifacts, the paper acknowledges that periodic artifacts are not entirely eliminated. This suggests room for further improvement in artifact reduction.

3) The method relies heavily on the pre-trained FLUX.1-dev model, which may limit its generalization to other domains or tasks. The paper does not explore how well the method performs when applied to different types of image degradation beyond the tested datasets.

**Questions For Authors:**

1) Could the authors provide more details on the scalability of FluxSR? For instance, how does the method perform on larger or more diverse datasets, and what are the implications for real-time applications?

2) Given that periodic artifacts are still present, are there any plans to further refine the model to address this issue? Could the authors explore additional regularization techniques or architectural changes to mitigate these artifacts?

3) While the paper compares FluxSR with other diffusion-based methods, it would be beneficial to include a more detailed comparison with non-diffusion-based approaches, especially in terms of computational efficiency and real-world applicability.

**Relation To Broader Scientific Literature:**

The paper builds on and advances existing research in image super-resolution, diffusion models, and flow matching theory. By introducing flow trajectory distillation (FTD), TV-LPIPS perceptual loss, and attention diversification loss (ADL), the authors address key challenges in real-world SR, such as computational efficiency, artifact reduction, and perceptual quality. The proposed FluxSR model achieves SOTA performance with only one diffusion step, providing a new direction for real-world SR research.

**Theoretical Claims:**

The paper provides a solid theoretical foundation for the proposed FluxSR model, supported by flow matching theory, mathematical formulations, and empirical evidence. The key innovations (FTD, TV-LPIPS perceptual loss, and ADL) are well-justified and contribute to the model's superior performance in real-world ISR. The theoretical claims are validated through extensive experiments, ablation studies, and visual comparisons, demonstrating the effectiveness of FluxSR.

---

> ### Author Rebuttal · Authors · 2025-04-01
>
> ## Response to Reviewer mAgS (denoted as R5)
>
> **Q5-1:** Could the authors provide more details on the scalability of FluxSR? For instance, how does the method perform on larger or more diverse datasets, and what are the implications for real-time applications?
>
> **A5-1:** We generate 10k noise-image pairs on the 220k-GPT4Vision-captions-from-LIVIS dataset as new training data. After training with a larger dataset, there are no significant changes in the quantitative results. However, we observe that the model trained with the larger dataset generated fewer high-frequency artifacts in the actual generated images. For real-time applications, further optimization, such as model pruning, quantization or more efficient techniques, would be helpful to improve real-time efficiency without sacrificing performance.
>
> **Q5-2:** Are there any plans to further refine the model to address this issue? Could the authors explore additional regularization techniques or architectural changes to mitigate these artifacts?
>
> **A5-2:** Yes, we increase the weights of the anti-artifact losses. Specifically, we increase the weight of TV-LPIPS to 2 and the weight of ADL to 0.2. In practice, we do not observe the high-frequency artifacts anymore, which demonstrates the effectiveness of the proposed anti-artifacts losses. We will include these results into the paper.
>
> **Q5-3:** While the paper compares FluxSR with other diffusion-based methods, it would be beneficial to include a more detailed comparison with non-diffusion-based approaches, especially in terms of computational efficiency and real-world applicability.
>
> **A5-3:** Thank you for your suggestion. We compare the computational efficiency and quantitative comparison of FluxSR with non-diffusion methods in the table below. Compared to non-diffusion methods, FluxSR produces significantly better visual results with higher image quality. Although FluxSR has higher computational complexity, it generates more realistic images. In highly degraded scenarios, if the user aims to generate high-quality images and is not too strict about inference speed, we believe FluxSR holds greater value.
>
> | Methods         | RealSR-JPEG | BSRGAN | ESRGAN | Real-ESRGAN | SwinIR | FeMaSR | FluxSR  |
> |-----------------|-------------|--------|--------|-------------|--------|--------|---------|
> | **Inference time / s** | 0.042       | 0.042  | 0.042  | 0.042       | 0.200  | 0.082  | 0.228   |
> | **MACs / T**    | 0.294       | 0.294  | 0.294  | 0.294       | 0.478  | 0.476  | 11.71   |
> | **# Params / B**| 0.017       | 0.017  | 0.017  | 0.017       | 0.027  | 0.033  | 11.99   |
>
> *Complex Analysis*
>
> | Method       | RealSR-JPEG | BSRGAN | ESRGAN | Real-ESRGAN | SwinIR | LDL   | FeMaSR | **FluxSR** |
> |--------------|-------------|--------|--------|-------------|--------|-------|--------|------------|
> | **MUSIQ**    | 50.54       | 65.58  | 42.37  | 63.22       | 63.82  | 63.22 | 64.88  | **70.75**  |
> | **MANIQA**   | 0.2927      | 0.3887 | 0.3100 | 0.3892      | 0.3818 | 0.3897| 0.4017 | **0.5495** |
> | **TOPIQ**    | 0.4118      | 0.5812 | 0.3864 | 0.5368      | 0.5306 | 0.5358| 0.5736 | **0.6670** |
> | **QAlign**   | 3.4416      | 3.9730 | 2.9680 | 4.0442      | 3.9661 | 4.0038| 4.0162 | **4.2134** |
>
> *RealSet65*
>
>
> **Q5-4:** The paper does not explore how well the method performs when applied to different types of image degradation beyond the tested datasets.
>
> **A5-4:** We evaluate FluxSR on face restoration task. Although no specific training set was used, our method still achieves good results. The specific quantitative results are shown in the table below.
>
> | Method           | RestoreFormer++ | VQFR   | CodeFormer | DAEFR  | PGDiff | DifFace | DiffBIR | OSEDiff | OSDFace | **FluxSR** |
> |------------------|-----------------|--------|------------|--------|--------|---------|---------|---------|---------|------------|
> | **MUSIQ**        | 71.484          | 70.906 | 74.001     | 72.698 | 68.599 | 65.116  | 72.272  | 69.322  | 73.935  | **75.908**  |
> | **MANIQA**       | 0.4902          | 0.4909 | 0.5034     | 0.4934 | 0.4460 | 0.4189  | 0.5839  | 0.4713  | 0.5162  | **0.6765**  |
> | **ClipIQA**      | 0.6950          | 0.6769 | 0.6918     | 0.6696 | 0.5653 | 0.5737  | 0.7441  | 0.6321  | 0.7106  | **0.7604**  |
>
> *WebPhoto-Test*
>
>
> | Method           | RestoreFormer++ | VQFR   | CodeFormer | DAEFR  | PGDiff | DifFace | DiffBIR | OSEDiff | OSDFace | **FluxSR** |
> |------------------|-----------------|--------|------------|--------|--------|---------|---------|---------|---------|------------|
> | **MUSIQ**        | 71.332          | 71.417 | 73.406     | 74.143 | 68.135 | 64.907  | 75.321  | 66.538  | 74.601  | **76.198**  |
> | **MANIQA**       | 0.4767          | 0.5044 | 0.4958     | 0.5205 | 0.4531 | 0.4299  | 0.6625  | 0.4616  | 0.5229  | **0.6665**  |
> | **ClipIQA**      | 0.7159          | 0.7069 | 0.6986     | 0.6975 | 0.5824 | 0.5924  | **0.8084** | 0.6235 | 0.7284  | 0.7847     |
>
> *Wider-Test*

---

### Official Review · Reviewer_jQYF · 2025-03-14

**Overall Recommendation:** 3

**Summary:**

The paper introduces FluxSR, a novel one-step diffusion model for Real-ISR (Real-World Image Super-Resolution). The primary goal is to reduce the high computational cost associated with multi-step diffusion models while preserving high-quality image generation. The key innovation is Flow Trajectory Distillation (FTD), which transfers the generative capabilities of a large-scale T2I diffusion model (FLUX.1-dev) into a single-step framework. Additionally, TV-LPIPS loss is introduced to suppress high-frequency artifacts, and Attention Diversification Loss (ADL) is used to prevent repetitive patterns.

**Claims And Evidence:**

1. The claims about FluxSR's performance improvements are well-supported by quantitative and visual results in Tables 1, 2, and Figure 5.
2. The claim that FTD prevents distribution shift is plausible but not directly validated by additional distribution analysis.
3. The effectiveness of TV-LPIPS is quantitatively supported but lacks a visual comparison.

**Essential References Not Discussed:**

A comparison to alternative single-step SR methods (e.g., GAN-based SR models like ESRGAN, Real-ESRGAN) would help contextualize the approach.

**Experimental Designs Or Analyses:**

1. The experimental setup is reasonable, using pre-generated noise-image pairs instead of real datasets.
2. The ablation study on loss functions is well-structured, but missing a visual comparison of TV-LPIPS.
3. A missing comparison of inference efficiency limits conclusions about computational benefits.

**Methods And Evaluation Criteria:**

1. The proposed FTD method is conceptually sound and effectively translates flow matching principles to super-resolution.
2. The evaluation criteria (PSNR, SSIM, LPIPS, DISTS, MUSIQ, MANIQA, TOPIQ, Q-Align) are appropriate for Real-ISR tasks.
3. However, adding inference speed and computational cost metrics would strengthen the evaluation.

**Other Comments Or Suggestions:**

1. Can you provide a quantitative comparison of inference time, MACs, and parameter count for FluxSR vs. other one-step and multi-step methods? This would help support the efficiency claims.
2. Can you provide a visual comparison of TV-LPIPS vs. LPIPS alone? This would help illustrate the effectiveness of the proposed perceptual loss.
3. How does the noise-to-image mapping in T2I models fundamentally differ from LR-to-SR degradations in practice? Could you provide a more detailed analysis to support the claim in Figure 2?
4. How sensitive is FluxSR to different training datasets? Would training on a different set of noise-image pairs alter its performance?
5. Can the proposed FTD method be extended to video super-resolution? Would additional temporal constraints be needed?

**Other Strengths And Weaknesses:**

Strengths:
1. The paper is well-written and clearly structured.
2. The mathematical formulation of FTD is well-integrated with flow-matching theory.
3. Introducing ADL for SR is a new contribution.

Weaknesses:
1. Some claims about the differences between T2I and SR mapping require stronger justification. The paper argues that the T2I noise-to-image mapping differs significantly from the LR-to-SR degradation process, necessitating FTD. However, in recent large-scale T2I models, the final stages of denoising already address degradations similar to LR-to-SR, making the proposed motivation less compelling.
A more thorough analysis (e.g., comparing distributions of features extracted from T2I and SR models) would strengthen this claim.
2. No quantitative evaluation of computational efficiency. The paper claims FluxSR achieves efficient inference, but there is no quantitative comparison of inference speed, MACs (Multiply-Accumulate Operations), or parameter count. A table comparing these metrics against one-step and multi-step baselines would clarify the trade-offs between computational efficiency and performance.
3. Lack of a visual ablation for TV-LPIPS. The ablation study in Table 4 demonstrates that TV-LPIPS improves perceptual quality, but a visual comparison (before and after applying TV-LPIPS) would provide a more intuitive understanding. Adding side-by-side images showing the effect of TV-LPIPS vs. LPIPS alone would strengthen the justification.

**Questions For Authors:**

None.

**Relation To Broader Scientific Literature:**

1. The paper builds on prior diffusion-based Real-ISR methods (e.g., OSEDiff, SinSR, TSD-SR) and flow matching models (e.g., ReFlow, InstaFlow).
2. The discussion of one-step vs. multi-step models is well-grounded in prior research.
3. The connection to large-scale T2I diffusion models (e.g., FLUX, SDXL) is relevant.

**Theoretical Claims:**

The Flow Trajectory Distillation formulation is mathematically consistent with flow matching theory. No formal proofs are included, but the derivations appear correct.

---

> ### Author Rebuttal · Authors · 2025-04-01
>
> ## Response to Reviewer jQYF (denoted as R4)
>
> **Q4-1:** How does the noise-to-image mapping in T2I models fundamentally differ from LR-to-SR degradations in practice? Could you provide a more detailed analysis to support the claim in Figure 2?
>
> **A4-1:** Although $x_t$ in the diffusion process *appears* to be similar to LR, they are fundamentally different. First, the distributions of degradations on these mappings are different. The calculation of $x_t$ is given by
>
> $$
> x_t = (1 - t) x_0 + t \epsilon,
> $$
>
> which is equivalent to adding Gaussian noise to the image. In contrast, LR is a real-world low-resolution image, which undergoes complex and unknown degradation. In other words, LR images may not be on the T2I trajectory, which is also a condition for the validity of Figure 2. Second, the degradations/noises are added in different domains. T2I adds noise in the latent space, but LR-to-SR can be regarded as adding degradations/noise onto the HR images. We highlight that these differences directly motivates us to reduce the distribution shift and propose our FTD method.
>
> **Q4-2:** Can you provide a quantitative comparison of inference time, MACs, and parameter count for FluxSR vs. other one-step and multi-step methods? This would help support the efficiency claims.
>
> **A4-2:** Yes, we have compared FluxSR's computational complexity with other multi-step and single-step methods, including inference speed, MACs, and parameter count. Despite using a 12B parameter model, the inference speed of our method is less than twice that of the fastest one-step diffusion ISR method.
>
> | Methods   | StableSR | DiffBIR | SeeSR | SUPIR | ResShift | SinSR | OSEDiff | TSD-SR | FluxSR |
> |--|--|--|--|--|--|--|--|---|--|
> | Inference step | 200 | 50 | 50 | 50 | 15 | 1 | 1 | 1 | 1 |
> | Inference time/s | 11.503 | 7.798 | 5.926 | 18.359 | 0.806 | 0.131 | 0.167 | 0.138 | 0.228 |
> | MACs/T | 75.81 | 24.52 | 32.34 | 120.41 | 4.90 | 2.09 | 2.27 | 2.91 | 11.71 |
> | # Params/B | 1.39 | 1.62 | 1.99 | 4.49 | 0.17 | 0.17 | 1.40 | 2.21 | 11.99 |
>
> **Q4-3:** Can you provide a visual comparison of TV-LPIPS vs. LPIPS alone? This would help illustrate the effectiveness of the proposed perceptual loss.
>
> **A4-3:** Thank you for your suggestion. We would like to show the qualitative comparison of the ablation studies. However, it is not allowed to upload any images according to the rules of rebuttal. In fact, our TV-LPIPS loss achieves better visual results than the LPIPS loss. As shown in Table 4 of the paper (the first two rows), this can be verified by the improved metrics that measure visual quality, including MUSIQ, ManIQA, and Q-Align. We will include more qualitative comparisons in the revised paper.
>
> **Q4-4:** How sensitive is FluxSR to different training datasets? Would training on a different set of noise-image pairs alter its performance?
>
> **A4-4:** We generate 10k noise-image pairs on the 220k-GPT4Vision-captions-from-LIVIS dataset as new training data. After training with a larger dataset, there are no significant changes in the quantitative results. However, we observe that the model trained with the larger dataset generated fewer high-frequency artifacts in the actual generated images.
>
> **Q4-5:** Can the proposed FTD method be extended to video super-resolution? Would additional temporal constraints be needed?
>
> **A4-5:**  Our FTD is indeed a promising approach for image SR, and we believe it could be extended to video SR with some adjustments. Incorporating temporal constraints (e.g., optical flow, temporal smoothness, feature alignment and propagation) between frames is crucial to avoid introducing flickering or artifacts across consecutive frames, and helps preserve motion consistency and smoothness. Due to the time limit, we leave it for future work.
>
> **Q4-6:** No Supplementary Material is provided.
>
> **A4-6:** In fact, we have provided supplementary material.
>
> **Q4-7:** A comparison to alternative single-step SR methods (e.g., GAN-based SR models like ESRGAN, Real-ESRGAN) would help contextualize the approach.
>
> **A4-7:** We have provided a comparison with non-diffusion methods in the supplementary material. Here, we present a comparison with BSRGAN, Real-ESRGAN, SwinIR, LDL, and FeMASR, as shown in the table below.
>
> | Method       | RealSR-JPEG | BSRGAN | ESRGAN | Real-ESRGAN | SwinIR | LDL   | FeMaSR | **FluxSR** |
> |--------------|-------------|--------|--------|-------------|--------|-------|--------|------------|
> | **MUSIQ**    | 50.54       | 65.58  | 42.37  | 63.22       | 63.82  | 63.22 | 64.88  | **70.75**  |
> | **MANIQA**   | 0.2927      | 0.3887 | 0.3100 | 0.3892      | 0.3818 | 0.3897| 0.4017 | **0.5495** |
> | **TOPIQ**    | 0.4118      | 0.5812 | 0.3864 | 0.5368      | 0.5306 | 0.5358| 0.5736 | **0.6670** |
> | **QAlign**   | 3.4416      | 3.9730 | 2.9680 | 4.0442      | 3.9661 | 4.0038| 4.0162 | **4.2134** |
>
> *RealSet65*

---

### Official Review · Reviewer_ozHQ · 2025-03-16

**Overall Recommendation:** 3

**Summary:**

This paper improves on the one-step diffusion-based super-resolution methods that target the real-world image super-resolution (Real-ISR) task by distilling on a larger and more advanced baseline image generation model (FLUX) compared to existing works that leverage Stable Diffusion as a backbone. It introduces Flow Trajectory Distillation (FTD) to address the distribution shift issue of existing methods. It also proposes using total variation as a perceptual loss and the ADL proposed by Guo et al. to emphasize restoring high-frequency details and improving generation quality.

## update after rebuttal

I appreciate the authors' additional experiments and justifications, which have adequately addressed my concerns. From my perspective, it is reasonable that the proposed method improves only on a subset of metrics, as different approaches naturally focus on different aspects of the problem. This is why, despite noting limitations such as lower performance on metrics like PSNR and the presence of over-smooth qualitative results, I initially recommended a rating of weak accept. Overall, I believe the paper's contributions outweigh its limitations, and I am inclined to maintain my recommendation of weak acceptance. Thanks

**Claims And Evidence:**

The TV-LPIPS component is claimed to emphasize the restoration of high-frequency components; however, in Figure 1, the proposed FluxSR method seems to generate over-smooth results that do not align with the original low-res image. For example, the helmet in the bottom row ignores all the high-frequency details that can be observed from the LR image.

Also, the ablation studies in Tables 3 and 4 only compare four of the eight metrics. Thus, it can be quite incomplete to show the effectiveness of each component. Besides, the reported PSNR is always inferior after incorporating this paper’s designs, which is detrimental to the justification of the claims.

**Essential References Not Discussed:**

N/A

**Experimental Designs Or Analyses:**

Overall, the experimental designs in this paper, such as the compared baselines, make sense to me. I acknowledge that there are many more one-step diffusion-based Real-ISR methods at the moment; however, I believe it is sufficient for the authors to only compare with the reported baselines.

**Methods And Evaluation Criteria:**

Overall, the evaluation criteria used in this paper, such as the metrics and benchmark datasets, make sense to me. This paper includes a range of metrics for evaluation. It adopts the standard RealESR-GAN’s degradation pipeline for creating the training data and DIV2K-val, RealSR, and RealSet65 as the test set, which, however, does not include the DrealSR that is widely used in various baselines.

**Other Comments Or Suggestions:**

In line 035 of the appendix, there is a question mask during referencing that needs to be fixed.

**Other Strengths And Weaknesses:**

Despite the practical contribution this paper makes, I am a bit concerned about the over-sharp qualitative results shown in the Appendix. Also, the inferior PSNR results shown in both the main Table and the Tables for ablation studies since these suggest that the super-resolved images can hallucinate some image details, which increases no-reference metrics, however, at the cost of the aforementioned full-reference metric like PSNR.

**Questions For Authors:**

Please check out the previous sections regarding my questions and concerns.

**Relation To Broader Scientific Literature:**

This paper mainly contributes to proposing the first one-step diffusion for Real-ISR based on a large model with over 12B parameters (FLUX.1-dev), highlighting its practical value.

**Theoretical Claims:**

The theoretical claims seem to make sense to me; however, I have not thoroughly checked or reproduced the derivations myself.

---

> ### Author Rebuttal · Authors · 2025-04-01
>
> ## Response to Reviewer ozHQ (denoted as R3)
>
> **Q3-1:** Does not include the DrealSR that is widely used in various baselines.
>
> **A3-1:** Thank you for pointing out this issue. The table below shows the quantitative comparison on the DRealSR dataset. From the table below, our FluxSR obtains significantly better results.
>
> | Method   | DiffBIR | SeeSR | ResShift | SinSR | OSEDiff | TSD-SR  | **FluxSR** |
> |-|-|-|-|-|-|-|-|
> | **PSNR**  | 25.91   | 28.35 | 26.42    | 27.33 | 24.20   | 25.93   | 25.92      |
> | **SSIM**  | 0.6190  | 0.8052| 0.7310   | 0.7237| 0.7355  | 0.7423  | 0.7592     |
> | **LPIPS** | 0.5347  | 0.3031| 0.4582   | 0.4444| 0.3429  | 0.3383  | 0.3418     |
> | **DISTS** | 0.2387  | 0.1665| 0.2382   | 0.2262| 0.1763  | 0.1708  | **0.1628**    |
> | **MUSIQ** | 36.18   | 34.51 | 30.52    | 32.79 | 37.22   | 36.18   | **37.82**  |
> | **MANIQA**| 0.5059  | 0.4736| 0.3018   | 0.3907| 0.4793  | 0.4272  | **0.5310** |
> | **QAlign**| 4.2402  | 4.2050| 4.2770   | 4.2704| 4.2503  | 4.2751  | **4.3356** |
>
> *DRealSR*
>
> **Q3-2:** Concern about the over-sharp qualitative results shown in the Appendix. Also, the cost of the aforementioned full-reference metric like PSNR.
>
> **A3-2:** We agree that our FTD is not very effective on PSNR, but the hallucination issue is also not severe since most images still retain high consistency with the LR image in terms of content. We highlight that PSNR does not always align well with human visual perception, which has been widely proved in existing works. Moreover, when the input images are severely degraded, the "hallucinated" details generated by our method are reasonable and realistic, and no-reference metrics are better at reflecting the perceptual quality of the images.
>
> **Q3-3:** In line 035 of the appendix, there is a question mask during referencing that needs to be fixed.
>
> **A3-3:** Thank you for pointing out this mistake. We will make the correction.
>
> **Q3-4:** In Figure 1, the proposed FluxSR method seems to generate over-smooth results that do not align with the original low-res image. For example, the helmet in the bottom row ignores all the high-frequency details that can be observed from the LR image.
>
> **A3-4:**  We agree that the generated helmet is smooth, but still visually reasonable. We highlight that our FluxSR is able to adaptively adjust the generative ability according to the risk of generating artifacts, showing better robustness than existing methods. For example, the texture of helmet is very blurry and suffers a high risk of artifacts. Existing methods consistently produce very poor results with severe distortions. Nevertheless, as for the nose and mouth in the face, since the textures can be easily imagined, our FluxSR works very well to produce detailed textures. Moreover, our FluxSR produces better visual details for text (see the second row of Figure 5).
>
> **Q3-5:** Also, the ablation studies in Tables 3 and 4 only compare four of the eight metrics. Thus, it can be quite incomplete to show the effectiveness of each component. Besides, the reported PSNR is always inferior after incorporating this paper’s designs, which is detrimental to the justification of the claims.
>
> **A3-5:** We expand the comparison to include additional metrics to provide a more complete evaluation of the performance, as shown in the table below. We highlight that our FTD consistently obtains better results on the metrics that measure visual quality. Regarding the PSNR results, it is important to note that PSNR is not always the best indicator of perceptual image quality, especially in severely degraded images. Our method focuses on enhancing perceptual fidelity and more visually realistic output, and it is better captured by no-reference metrics. It may not always align with the traditional PSNR metric, but it contributes positively to the overall perceptual quality. We will clarify this trade-off in the revised manuscript, emphasizing that our method is optimized for visual realism rather than solely maximizing PSNR.
>
> | Method        | **PSNR** | **SSIM** | **LPIPS** | **DISTS** | **MUSIQ** | **MANIQA** | **TOPIQ** | **Q-Align** |
> |--|--|--|--|--|--|--|--|--|
> | w/o FTD       | **26.33** | **0.7580**   | 0.3801    | 0.2200    | 56.02     | 0.3775     | 0.4006    | 3.5170      |
> | FTD (ours)    | 24.67    | 0.7133   | **0.3324**| **0.1896**| **67.84** | **0.5203** | **0.6530**| **4.1473**  |
>
> *Ablation study on FTD*
>
> | $\mathcal{L}_{\text{LPIPS}}$ | $\mathcal{L}_{\text{TV-LPIPS}}$ | $\mathcal{L}_{\text{EA-DISTS}}$ | $\mathcal{L}_{\text{ADL}}$ | **SSIM** | **LPIPS** | **DISTS** | **TOPIQ** |
> |--|--|--|--|--|--|--|--|
> | ✓ |  |  |  | 0.6893   | 0.3459    | 0.2096    | 0.6242    |
> |  | ✓ |  |  | 0.6999   | 0.3369    | 0.1933    | 0.6387    |
> |  |  | ✓ |  | 0.7283   | 0.3423    | 0.1970    | 0.6400    |
> |  |  | ✓ | ✓ | **0.7339**| 0.3332    | 0.1915    | 0.6427    |
> | ✓ |  | ✓ | ✓ | 0.7133   | **0.3324**| **0.1896**| **0.6530**|
>
> *Ablation study on different loss functions*

---

### Official Review · Reviewer_uJUz · 2025-03-16

**Overall Recommendation:** 3

**Summary:**

The authors claim that most existing one-step diffusion methods are constrained by the performance of the teacher model, where poor teacher performance results in image artifacts. To this end, the authors proposed a one-step diffusion Real-ISR technique, namely FluxSR, based on FLUX.1-dev and flow matching models. The authors introduce Flow Trajectory Distillation (FTD) to distill a one-step model from the teacher model. The author provides comparative experiments with the state-of-the-art ISR methods.


## update after rebuttal
The authors have addressed the majority of my concerns. Accordingly, I have updated my score to 3: Weak Accept.

**Claims And Evidence:**

The author's claims are clear, but the evidence to support these claims is insufficient. There are mainly the following problems:

(1) The authors analyse the possible negative outcomes of VSD or GANs in Sec. 4.1, but lack experimental support.

**Essential References Not Discussed:**

None

**Experimental Designs Or Analyses:**

I have carefully checked the soundness/validity of any experimental designs and analyses, and there are the following problems:

(1) The outstanding performance of the proposed method may be mainly attributed to the FLUX model. If replaced with other baselines, the performance gain brought by the proposed pipeline in the paper is questionable.

(2) The FTD proposed by the authors introduces the SR flow trajectory based on the existing T2I flow trajectory distillation, but lacks quantitative and qualitative analysis to verify the effectiveness of this improvement.

(3) The authors proposed a large model friendly training strategy. What advantages does it bring to model training and inference? The authors should report this in the experimental analysis.

(4) There is a lack of corresponding comparison and analysis of model inference speed in the experiment.

(5) The paper lacks a comparison with the latest SUPIR [1]. An analysis of their performance and efficiency is necessary in the paper.

(6) The ablation study in the paper lacks a qualitative comparison.

[1] Scaling up to excellence: Practicing model scaling for photo-realistic image restoration in the wild, CVPR 2024

**Methods And Evaluation Criteria:**

The proposed method and evaluation datasets used by the authors are reasonable.

**Other Comments Or Suggestions:**

The color markings in Tab.1 and Tab.2 of the paper are confusing. There is no specific explanation of what red bold and blue bold mean. What is the difference between them and bold?

**Other Strengths And Weaknesses:**

The paper lacks innovation. The TV-LPIPS and ADL proposed in the paper are both existing works. The outstanding performance of the proposed method may be mainly attributed to the introduced FLUX model.

**Questions For Authors:**

(1) The authors analyse the possible negative outcomes of VSD or GANs in Sec. 4.1, but lack experimental support. The author should support this conclusion with corresponding analysis.

(2) The FTD proposed by the authors introduces the SR flow trajectory based on the existing T2I flow trajectory distillation, but lacks quantitative and qualitative analysis to verify the effectiveness of this improvement. Corresponding experimental analysis should be provided.

(3) Is the outstanding performance of the proposed method mainly attributed to the FLUX model? If it is replaced with other baselines, will there still be performance advantages? Further analysis should be provided.

(4) There is a lack of corresponding comparison and analysis of model inference speed in the experiment. The author should compare the inference speed of the proposed FluxSR with other methods.

(5) The authors proposed a large model friendly training strategy. What advantages does it bring to model training and inference? The authors should report this in the experimental analysis.

(6) The color markings in Tab.1 and Tab.2 of the paper are confusing. There is no specific explanation of what red bold and blue bold mean. What is the difference between them and bold?

(7) The paper lacks a comparison with the latest SUPIR [1]. An analysis of their performance and efficiency is necessary in the paper.

(8) The ablation study in the paper lacks a qualitative comparison. Corresponding comparisons should be provided.

[1] Scaling up to excellence: Practicing model scaling for photo-realistic image restoration in the wild, CVPR 2024

If the author could address these issues, I would be inclined to raise my score.

**Relation To Broader Scientific Literature:**

This paper implements the Real-ISR task based on existing methods (FLUX model, TV-LPIPS, ADL), and further proposes an improved strategy (FTD). The key idea of FTD is introducing the LR-to-HR flow in SR based on the flow matching theory.

**Theoretical Claims:**

I have carefully checked the correctness of the proofs for theoretical claims and found no relevant problems.

---

> ### Author Rebuttal · Authors · 2025-04-01
>
> ## Response to Reviewer uJUz (denoted as R2)
>
> **Q2-1:** The paper lacks innovation. The TV-LPIPS and ADL proposed in the paper are both existing works.
>
> **A2-1:** The main contribution of this paper is the introduction of FTD, and our method is the first work to distill a large-scale flow matching model like FLUX into a one-step model for image super-resolution. Existing one-step methods struggle with large diffusion models. Specifically, VSD (with optional GAN loss) requires two additional copies of the large model in GPU memory, exceeding the capacity of an 80GB A800 GPU. Our FTD is large-model-friendly, requiring only about 55GB of GPU memory per card for training, and 23.7GB for inference with 512px resolution.
>
> **Q2-2:** The authors analyze the possible negative outcomes of VSD or GANs in Sec. 4.1, but lack experimental support.
>
> **A2-2:** We add experiments to support our argument. Specifically, we generate images using teacher models of OSEDiff, AddSR, FluxSR and compute the FID between original T2I images and SR images. FluxSR’s FID is significantly lower than OSEDiff and AddSR, indicating that FTD avoids distribution shift.
>
> | Method   | AddSR | OSEDiff | **FluxSR** |
> |-|-|-|-|
> | **FID**  | 43.9  | 43.0 | **34.9**  |
>
> **Q2-3:** The FTD lacks quantitative and qualitative analysis to verify the effectiveness of this improvement.
>
> **A2-3:** We have presented quantitative results of the FTD ablation in the ablation study section, demonstrating its effectiveness. We will include qualitative comparisons in the supplementary material.
>
> **Q2-4:** Is the outstanding performance of the proposed method mainly attributed to the FLUX model?
>
> **A2-4:** We replace FluxSR's baseline with SD3-medium and retrained it, comparing it with TSD-SR that is also trained on SD3-medium . The results show that FluxSR outperforms TSD-SR, indicating that the performance improvement comes from our method.
>
> | Method | TSD-SR | FluxSR on SD3 |
> |-|-|-|
> | MANIQA | 0.4884 | **0.5310** |
> | TOPIQ  | 0.6526 | **0.6603** |
> | QAlign | 4.2258 | **4.3422** |
>
> **Q2-5:** Lack of corresponding comparison and analysis of model inference speed.
>
> **A2-5:** We compare FluxSR’s computational complexity with other multi-step and one-step methods. Despite using a 12B parameter model, the inference time of FluxSR is no more than 0.1s longer than the current fastest method.
>
> | Methods   | StableSR | DiffBIR | SeeSR | SUPIR | ResShift | SinSR | OSEDiff | TSD-SR | FluxSR |
> |-|-|-|-|-|-|-|-|-|-|
> | Inference step | 200 | 50 | 50 | 50 | 15 | 1 | 1 | 1 | 1 |
> | Inference time/s | 11.503 | 7.798 | 5.926 | 18.359 | 0.806 | 0.131 | 0.167 | 0.138 | 0.228 |
> | MACs/T | 75.81 | 24.52 | 32.34 | 120.41 | 4.90 | 2.09 | 2.27 | 2.91 | 11.71 |
> | # Params/B | 1.39 | 1.62 | 1.99 | 4.49 | 0.17 | 0.17 | 1.40 | 2.21 | 11.99 |
>
> **Q2-6:** What advantages does Large-Model-Friendly Training bring to model training and inference?
>
> **A2-6:** The Large-Model-Friendly Training (LMFT) strategy reduces memory usage and training time. During inference, it reduces memory usage and inference latency.
>
> | Method   | Training CUDA usage | Training time (per iteration) | Inference CUDA usage | Inference time (512px) |
> |-|-|-|-|-|
> | w/o LMFT | 76.2GB | 6.91s | 34.82GB | 457ms |
> | w LMFT   | 55.4GB | 4.43s | 23.77GB | 228ms |
>
> **Q2-7:** The color markings in Tab.1 and Tab.2 of the paper are confusing.
>
> **A2-7:** In Tables 1 and 2, red and blue bold represent the best and second-best methods among all approaches. The best method in one-step diffusion will also be bolded separately. Red bold indicates it is the best overall, while blue bold indicates it is the best in one-step methods but second-best overall.
>
> **Q2-8:** Lacks a comparison with SUPIR.
>
> **A2-8:** We add a comparison with SUPIR on RealLQ250 datasets in the table below. And we have included an efficiency comparison with SUPIR in **A5**.
>
> | Method      | DiffBIR | SeeSR  | SUPIR  | ResShift | SinSR  | AddSR  | OSEDiff | TSD-SR  | **FluxSR** |
> |-|-|-|-|-|-|-|-|-|-|
> | MUSIQ | 71.61   | 70.53  | 65.91  | 59.45    | 65.38  | 64.23  | 69.56   | 72.10   | **72.65**  |
> | MANIQA | 0.5472  | 0.4971 | 0.3907 | 0.3383   | 0.4264 | 0.3707 | 0.4230  | 0.4596  | **0.5490** |
> | TOPIQ | 0.6835  | 0.6653 | 0.5631 | 0.4709   | 0.5790 | 0.5470 | 0.6075  | 0.6456  | **0.6848** |
> | QAlign | 4.2307  | 4.1652 | 4.1442 | 3.6340   | 3.7426 | 3.8884 | 4.2484  | 4.1682  | **4.4077** |
>
> *RealLQ250*
>
> **Q2-9:** The ablation study in the paper lacks a qualitative comparison.
>
> **A2-9:** We would like to show the qualitative comparison of the ablation studies. However, it is not allowed to upload any images according to the rules of rebuttal. However, the metrics in our paper (MUSIQ, ManIQA, Q-Align) reflect image quality, showing that our method outperforms others in visual results, with TV-LPIPS and ADL effectively reducing high-frequency artifacts. We will include more qualitative comparisons in the revised paper.

---

### Official Review · Reviewer_m3JT · 2025-03-17

**Overall Recommendation:** 3

**Summary:**

This paper proposes FluxSR, a one-step diffusion model for real-world image super-resolution (Real-ISR). The author introduces Flow Trajectory Distillation (FTD) to distill multi-step diffusion models into a single step. FluxSR addresses distribution shifts by aligning noise-to-image and low-to-high-resolution flow trajectories. The method also introduces TV-LPIPS and Attention Diversification Loss (ADL) to reduce artifacts.

**Claims And Evidence:**

Most claims in the paper are clearly supported by convincing experimental results, particularly regarding the effectiveness of Flow Trajectory Distillation (FTD) for aligning flow trajectories and improving realism. However, the claim related to the reason why high-frequency artifacts emerge ("high-frequency artifacts due to token similarity in transformers") is not sufficiently explained or supported.

**Essential References Not Discussed:**

I did not find any critical related works missing from the paper. The authors cited and discussed the relevant prior methods and ideas necessary for understanding their key contributions.

**Experimental Designs Or Analyses:**

I checked the soundness and validity of the experimental designs, particularly the ablation studies, which are comprehensive and effectively demonstrate the contribution of each proposed component (FTD, TV-LPIPS, and ADL). However, one important shortcoming is the lack of visualization results in these ablation studies. Specifically, visual comparisons illustrating how TV-LPIPS and ADL mitigate high-frequency artifacts would significantly strengthen and clarify the analysis.

**Methods And Evaluation Criteria:**

The proposed method (FluxSR) is reasonable and effective in one-step diffusion methods for Real-ISR tasks. The evaluation metrics chosen by the authors are appropriate and cover both perceptual and fidelity aspects comprehensively.

However, the experimental evaluation is somewhat limited due to the use of three datasets, of which DIV2K is synthetic and does not fully reflect real-world complexities. Evaluating on additional, diverse real-world datasets would further validate the generalizability and robustness of the proposed method.

**Other Comments Or Suggestions:**

1. There are some inconsistencies and typos regarding mathematical notations. For example, the formulation presented in Algorithm 1 (line 9) is inconsistent with the corresponding equation shown in Figure 3. Clarifying and correcting these discrepancies would improve readability.
2. Although Flux is described as a DiT-based diffusion model, Figure 3 visually depicts it as a U-Net structure, potentially misleading readers. Revising the figure to accurately represent Flux’s architecture (DiT-based transformer) would avoid confusion.

**Other Strengths And Weaknesses:**

Strengths:
1. The paper introduces an effective approach (Flow Trajectory Distillation) that addresses the limitations (e.g., distribution shift) of existing one-step diffusion methods.
2. The paper is well-organized and generally clear, especially in terms of theoretical derivations and method explanations.

Weaknesses:
1. Incomplete Ablation on FTD: A key contribution of FTD is preserving the prior knowledge from the powerful teacher model (Flux). However, the authors only provide an ablation using the reconstruction loss at the single step ($T_L$). It remains unclear whether training the full trajectory ($[T_L , 1]$) using only reconstruction loss (without FTD) could achieve comparable results. Such an ablation would be crucial for validating the specific advantage provided by FTD, but is currently missing.
2. Insufficient Explanation for Artifacts: The authors do not adequately explain why artifacts occur. If the final distribution aligns closely with Flux’s HR distribution, these artifacts should theoretically not emerge. The current hypothesis attributing artifacts to token similarity lacks empirical validation. Providing visualization of attention maps or investigating whether artifacts result from insufficient training time or limited training data would clarify this issue.
3. Limited Evaluation on Real Datasets: The paper evaluates mainly on two datasets, with one (DIV2K) relying on synthetic degradations, limiting the strength of the empirical results. Evaluations on larger and more diverse real-world datasets (e.g., RealLR200 in SeeSR or RealLQ250 in DreamClear) would enhance the robustness and credibility of the experimental findings.

**Questions For Authors:**

See the weaknesses.

**Relation To Broader Scientific Literature:**

The proposed method, FluxSR, builds upon recent advances in one-step diffusion models (e.g., OSEDiff, TSD-SR), addressing their common limitations related to distribution shifts between teacher and student models through Flow Trajectory Distillation (FTD). Additionally, the proposed loss functions (TV-LPIPS and ADL) closely relate to perceptual loss-based super-resolution approaches (like RealESRGAN).

**Theoretical Claims:**

The paper contains relatively few theoretical claims, primarily focused on clearly defined equations for Flow Trajectory Distillation (FTD). I have checked the main theoretical derivations and formulations provided (e.g., Equations 10–18). These derivations are straightforward and clear. I did not find any significant issues.

---

> ### Author Rebuttal · Authors · 2025-04-01
>
> ## Response to Reviewer m3JT (denoted as R1)
> We sincerely thank the reviewer for the constructive comments. We provide the detailed responses to all the concerns below.
>
> **Q1-1:** Incomplete Ablation on FTD.
>
> **A1-1:** We add the relevant experiments. The results are shown in the table below. In practice, training the entire flow trajectory on $[T_L,1]$ yields similar results with training on a single time step. The main reason is that it is extremely hard to enforce a one-step model to directly mimic the entire flow trajectory, without explicitly preserving the generative ability of the teacher model. Instead, our FTD obtains significantly better results in most metrics that are widely used to evaluate visual quality.
>
> | Method               | PSNR  | MUSIQ | MANIQA | Q-Align |
> |----------------------|-------|-------|--------|---------|
> | $L_{rec}$ in $T_L$    | **26.33** | 56.02 | 0.3775 | 3.5170  |
> | $L_{rec}$ in $[T_L,1]$| 25.94 | 55.63 | 0.3900 | 3.6993  |
> | $L_{rec} + FTD$       | 24.67 | **67.84** | **0.5203** | **4.1473**  |
>
> *Ablation study on FTD*
>
> **Q1-2:** Insufficient Explanation for Artifacts.
>
> **A1-2:** To investigate the artifact issue, we visualize the attention/feature maps and found that the similarity between tokens is quite high, and there are indeed repeated features in some dimensions. Additionally, we observe that using the official FLUX model for one-step inference also results in high-frequency artifacts, which we believe is a characteristic of FLUX itself. When using only the reconstruction loss, we observed even more severe periodic artifacts, so these artifacts are not caused by FTD. We hypothesize that this issue is mainly attributed to the limited training data, which are significantly smaller than the ones used to train the multi-step FLUX. To verify this, when using a larger dataset, such as 10k noise-image pairs generated from the 220k-GPT4Vision-captions-from-LIVIS dataset, we observed a clear reduction in periodic artifacts.
>
> **Q1-3:** Limited Evaluation on Real Datasets.
>
> **A1-3:** According to your suggestions, we add the comparison on these two datasets, and the results are shown in the table below. In fact, FluxSR achieves the best results. We will include these into the paper.
>
> | Method      | DiffBIR | SeeSR  | SUPIR  | ResShift | SinSR  | AddSR  | OSEDiff | TSD-SR  | **FluxSR** |
> |-------------|---------|--------|--------|----------|--------|--------|---------|---------|------------|
> | MUSIQ       | 71.61   | 70.53  | 65.91  | 59.45    | 65.38  | 64.23  | 69.56   | 72.10   | **72.65**  |
> | MANIQA      | 0.5472  | 0.4971 | 0.3907 | 0.3383   | 0.4264 | 0.3707 | 0.4230  | 0.4596  | **0.5490** |
> | TOPIQ       | 0.6835  | 0.6653 | 0.5631 | 0.4709   | 0.5790 | 0.5470 | 0.6075  | 0.6456  | **0.6848** |
> | QAlign      | 4.2307  | 4.1652 | 4.1442 | 3.6340   | 3.7426 | 3.8884 | 4.2484  | 4.1682  | **4.4077** |
>
> *RealLQ250*
>
> | Method      | DiffBIR | SeeSR  | SUPIR  | ResShift | SinSR  | OSEDiff | AddSR  | TSD-SR  | **FluxSR** |
> |-------------|---------|--------|--------|----------|--------|---------|--------|---------|------------|
> | MUSIQ       | 69.63   | 69.75  | 64.88  | 59.87    | 65.11  | 65.42   | 69.61  | 71.02   | **71.60**  |
> | MANIQA      | 0.5526  | 0.5045 | 0.4677 | 0.3591   | 0.4549 | 0.3945  | 0.4388 | 0.4884  | **0.5588** |
> | TOPIQ       | 0.6772  | 0.6635 | 0.5870 | 0.4990   | 0.5998 | 0.5634  | 0.6083 | 0.6526  | **0.6814** |
> | QAlign      | 4.2529  | 4.2399 | 4.1675 | 3.7959   | 3.9317 | 4.0156  | 4.2895 | 4.2258  | **4.4004** |
>
> *RealLR200*
>
> **Q1-4:** There are some inconsistencies and typos regarding mathematical notations. For example, the formulation presented in Algorithm 1 (line 9) is inconsistent with the corresponding equation shown in Figure 3. Clarifying and correcting these discrepancies would improve readability.
>
> **A1-4:** Thank you for noticing and pointing out this mistake. We did make an error while editing the image. Specifically, the formula in Figure 3 corresponding to line 9 of Algorithm 1 should have a "$+$" sign instead of a "$-$". We will correct this error.
>
> **Q1-5:** Although Flux is described as a DiT-based diffusion model, Figure 3 visually depicts it as a U-Net structure, potentially misleading readers. Revising the figure to accurately represent Flux’s architecture (DiT-based transformer) would avoid confusion.
>
> **A1-5:** Thank you for pointing this out. We will delete the U-Net structure and replace it with the DiT architecture in Figure 3.

---

### Decision · Program_Chairs · 2025-05-01

**Decision:**

Accept (poster)

**Comment:**

This paper proposes a one-step diffusion method based on flow trajectory distillation for image super-resolution. The paper originally received 2xWeakReject and 3xWeakAccept. The main concerns include insufficient explanations, limited evaluations and novelty, lack of discussions on computational efficiency, etc. The authors have provided rebuttals and addressed most concerns of reviewers. Afterward, two reviewers raise their ratings and all reviewers lean to accept this paper. The authors are suggested to carefully revise the paper and incorporate newly conducted experiments according to the comments and discussions.